# Characterization of A π−π stacking cocrystal of 4-nitrophthalonitrile directed toward application in photocatalysis

Ting Xue[1], Cheng Ma[2], Le Liu[1], Chunhui Xiao[1]✉, Shao-Fei Ni[2]✉ & Rong Zeng[1]✉

Photoexcitation of the electron-donor-acceptor complexes have been an effective approach to achieve radicals by triggering electron transfer. However, the catalytic version of electron-donor-acceptor complex photoactivation is quite underdeveloped comparing to the well-established utilization of electronically biased partners. In this work, we utilize 4-nitrophthalonitrile as an electron acceptor to facilitate the efficient π-stacking with electron-rich aromatics to form electron-donor-acceptor complex. The characterization and energy profiles on the cocrystal of 4-nitrophthalonitrile and 1,3,5-trimethoxybenzene disclose that the electron transfer is highly favorable under the light irradiation. This electron acceptor catalyst can be efficiently applied in the benzylic C−H bond photoactivation by developing the Giese reaction of alkylanisoles and the oxidation of the benzyl alcohols. A broad scope of electron-rich aromatics can be tolerated and a mechanism is also proposed. Moreover, the corresponding π-anion interaction of 4-nitrophthalonitrile with potassium formate can further facilitate the hydrocarboxylation of alkenes efficiently.

Over the past decades, spontaneous aggregates of a pair of electron rich and poor molecules, known as electron-donor-acceptor (EDA) complexes, have drawn photochemists' significant attentions since they possess new visible-light absorptions to facilitate the excitation of the poor-absorption compounds[1–5]. The weak noncovalent interactions between the donor and acceptor moiety might enable the new molecular orbitals formation, and upon irradiation, the intramolecular single electron transfer (SET) would trigger radical reactions under mild conditions (Fig. 1a)[6,7]. Particularly, the distinctly catalytic EDA complex photochemistry would be achieved by using a catalytic partner, dramatically expanding the synthetic potentials of this strategy[8].

Despite the fact that much attention had been paid, these EDA complex catalysis majorly focused on the use of the electron-rich catalysts for conversion of the electron-poor substrates adorned with

a proper leaving group[9–17]. In contrary, such reports on the reactions using electron-poor catalysts (electron acceptor) are quite limited (Fig. 1b)[8] In 2020, Ooi[18]. reported the single electron transfer between $B(C_6F_5)_3$ and N,N-dialkylanilines. The π-orbital interaction was a key intermediate for the visible-light absorption, oxidation of N-atom for the α-aminoalkyl radical generation, and subsequent radical addition for achieving the α-alkylation. This aniline oxidation strategy could be further expanded by Tang[19] and Sundén[20] to develop aerobic conversions of indoles and N,N-dialkylanilines using catalytical $B(C_6F_5)_3$ and dibenzoylethylene, respectively. Alternatively, Melchiorre[21] pioneeringly utilized tetrachlorophthalimides as catalytic acceptors to photoactivate the 1,4-dihydropyridine (DHP), alkylsilicates, and organotrifluoroborates. The single electron oxidation and then fragmenting of the redox auxiliary (RA) facilitated the reactive radical formation after single electron transfer. The EDA complex

[1]School of Chemistry, Xi'an Jiaotong University, Xi'an 710049, PR China. [2]Department of Chemistry and Key Laboratory for Preparation and Application of Ordered Structural Materials of Guangdong, Shantou University, Shantou 515063 Guangdong, PR China. ✉e-mail: chunhuixiao@xjtu.edu.cn; sfni@stu.edu.cn; rongzeng@xjtu.edu.cn

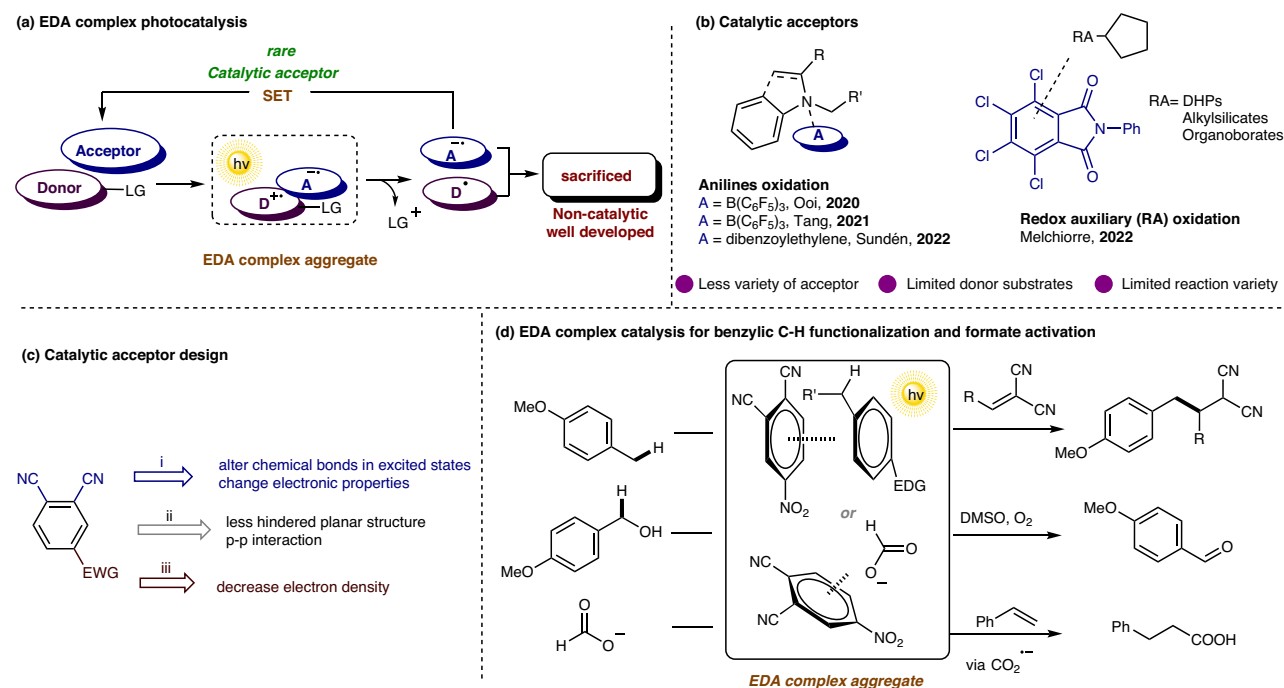

**Fig. 1 | EDA complex photoactivation. a** EDA complex photocatalysis. **b** Catalytic acceptors for EDA complex photoactivation. **c** The design of a catalytic acceptor. **d** EDA complex catalysis in this work.

photochemistry is still highly desired to develop protocols using more varieties of acceptor catalysts as well as electron donor substrates (beyond arylamines or R − RA) in order to achieve diverse transformations, furthermore, it should be noted that the direct single electron oxidation of arenes and subsequent benzylic C−H bond functionalization has been rarely realized via EDA complex catalysis[22].

We envisioned that an electron-withdrawing group substituted phthalonitrile, such as nitrophthalonitrile, would be suitable acceptor catalyst via π−π stacking to oxidize electron-rich arenes (Fig. 1c). The planar structure presents very small steric hindrance in the vertical axis, offering great opportunity for π−π interactions[14]. In addition, the electron-withdrawing group as well as two cyano groups would further decrease the electron density of the aromatic ring, enhancing the electron accepting ability. Herein, we examine this hypothesis and report the characterization of an electron-donor-acceptor complex of 4-nitrophthalonitrile with electron-rich (e-rich) arenes and the use of 4-nitrophthalonitrile as a proper acceptor catalyst for benzylic C−H photoactivation. This catalyst could further realize the formate activation via π-anion interaction (Fig. 1d).

## Results
### Characterization of A π−π stacking cocrystal
We started our exploration by preparing the photoactive EDA complex with an e-rich arene. Specifically, a yellow crystal was obtained by slow evaporation in the acetone solution of 1,3,5-trimethoxybenzen and 4-nitrophthalonitrile, which was assigned as a cocrystal in 1:1 ratio (Fig. 2a). 4-Nitrophthalonitrile and 1,3,5-trimethoxybenzene were packing face to face aligned along *a*-axis. In addition, the average distance between the plane of 4-nitrophthalonitrile and 1,3,5-trimethoxybenzene was 3.28 Å (Fig. 2a), which was in the range of inner-sphere EDA complex ($r_{DA} \approx 3.1 \pm 0.2$ Å)[3] Although none of two compounds can absorb visible light themselves, this cocrystal exhibits strong absorption at the $\lambda = 389 \sim 457$ nm (Fig. 2b). Moreover, DFT calculations on the orbitals of crystal structure disclosed that the HOMO or LUMO orbitals were distributed to the aromatic rings of 1,3,5-trimethoxybenzene or 4-nitrophthalonitrile, respectively (Fig. 2c). Furthermore, the electron-hole distribution of the EDA aggregate further supported the strong

interaction between two molecules, and upon irradiation, the degree of the electron transfer between donor and acceptor is about 0.984 (Fig. 2d).

Moreover, the photoinduced electron transfer in the cocrystal of 4-nitrophthalonitrile and 1,3,5-trimethoxybenzene could be directly detected by electron paramagnetic resonance (EPR) spectroscopy (Fig. 2e). Unsurprisingly, no radical signal was recorded without irradiation. Once the cocrystal was irradiated with 390 nm light (ex situ of the magnet), a build-up of a radical signal, assigned as a carbon radical ($g = 2.003$), could be observed in the EPR spectrum, proving an electron transfer process between 4-nitrophthalonitrile and 1,3,5-trimethoxybenzene initiated by irradiation. When the saturated solution of 4-nitrophthalonitrile and 1,3,5-trimethoxybenzene in DMSO was examined under irradiation, a similar signal could be obtained, albeit in a relatively low resolution, probably due to the low concentration of radical species. Delightedly, a much higher resolution signal could be obtained by switching the solvent to DMF. These results all indicated that the electron transfer between 4-nitrophthalonitrile and 1,3,5-trimethoxybenzene might be induced by light to general carbon radical species.

### Optimization of reaction conditions
These encouragingly preliminary results on efficient electron transfer had driven us to examine the catalytic benzylic C − H functionalization of toluene derivatives via the EDA complex catalysis[23–25]. Gladly, the reaction of 4-methylanisole (**1a**) with benzylidenemalononitrile (**2a**) could indeed happen smoothly under irradiation of 390 nm LEDs by simply using 4-nitrophthalonitrile **cat1** as photocatalyst, obtaining the C − H alkylation product **3aa** in 75% yield (71% isolated yield) (Fig. 3). The removal of either one cyano group (**cat2**) or the nitro group (**cat3**-**cat5**) from the catalyst showed lower efficiencies, obtaining the desired product in 13-60%. Notably, 27% of the desired product could be formed in the absence of **cat1** (Supplementary Table 1 in SI), indicating that **cat3-cat5** might be not true catalysts and the reaction could be driven by photo directly in lower efficiency. Interestingly as comparations, the well-developed photosensitizers, such as 4CzIPN (**cat6**) and pyrylium tetrafluoroborate (**cat7**), presented relatively lower

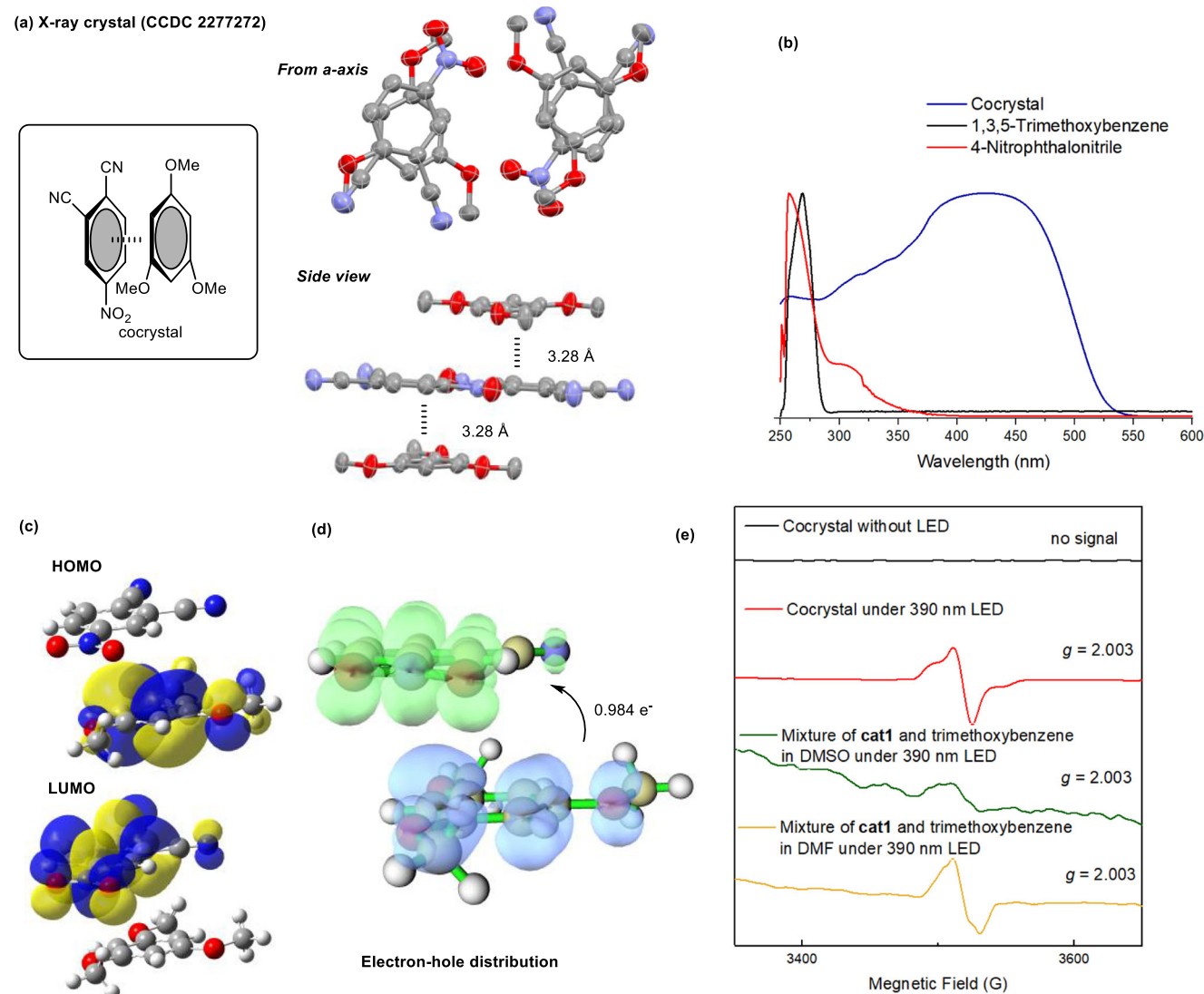

**Fig. 2 | Characterization and property of the cocrystal. a** Packing structure of the cocrystal of 4-nitrophthalonitrile and 1,3,5-trimethoxybenzene. **b** The UV-Vis absorption spectra of 4-nitrophthalonitrile, 1,3,5-trimethoxybenzene and their cocrystal. **c** DFT calculation on the HOMO and LUMO orbitals of the crystal structure. **d** DFT calculation on the electron-hole distribution of single complex in crystal structure. **e** EPR monitoring experiments of the cocrystal and solution of 4-nitrophthalonitrile and 1,3,5-trimethoxybenzene in the absence or presence of 390 nm light.

reactivities, while only the extremely e-poor catalyst (**cat7**) can give the promising result (59%). Not surprisingly, the electron-donated photosensitizers, such as Eosin Y (**cat8**) and its disodium salt (**cat9**), failed to produce the desired product, proving the crucial role of the electron hole.

## Substrate scopes

With this acceptor catalyst in hand, the applicable scope of electron-rich aromatics has been extensively tested (Fig. 4). A broad range of electron-rich methylarenes readily underwent alkylation with **2a** in moderate to excellent yields. The substitution position of methoxyl group had a significant influence on the C−H bond activation, since substrates with a methoxyl group at the *para*-position (**1a**) or *ortho*-position (**1b**) achieved dramatically different results (71% *vs* 16%). 3-Methylanisole (Supplementary Table 4, **1ad**) was not applicable to the catalytic system, and no product was observed. The carbon-halogen bonds, such as C−F, C−Cl, C−Br, and C−I bond, could be all tolerated and remained during the transformations, albeit in only 31−55% yields. Interestingly, when anisole derivatives containing two methyl groups, such as 2,4-dimethylanisole and 3,4-dimethylanisole,

were used, the C−H alkylation would happen highly selectively at the *para*-position of the methoxy group. Di-(4-tolyl)ether produced the mono-alkylation product in 46% yield. The electronic interaction of the catalyst with more e-rich arenes, such as 2,6-dimethoxytoluene **1k** and 2,4,6-trimethoxytoluene **1l**, might be enhanced significantly, resulting in the yield increasing to quantitative and 94%, respectively. Notably, that the steric hindrance of the 2,6-disubstitution did not affect the efficiencies suggested the C−H bond cleavage be not driven by the intermolecular hydrogen atom abstraction. In addition, electron-rich heteroarenes, such as thiophene (**1m**), furan (**1n**), benzofuran (**1o-p**), benzothiophene (**1q-r**), could be tolerated well, undergoing the C−H bonds alkylation in moderate to good yields.

Next, secondary benzylic C(sp³)−H bond alkylations were examined (Fig. 4). The existence of benzylic substituents might hinder the interaction with the catalyst and decrease reaction efficiencies, therefore, although desired products were formed successfully, these reactions typically had to be prolonged to 72 hours to achieve the higher conversions and yields. The reaction of 4-ethylanisole smoothly produced the desired product, as a pair of diastereomers (*dr* = 1:2.2), in 87% yield. A series of diverse functional groups, such as -vinyl (**1t**),

**Fig. 3 | Screening of the photocatalysts.** Reactions were conducted with **1a** (0.4 mmol), **2a** (0.2 mmol), 4-nitrophthalonitrile (0.02 mmol) in DMSO (2 mL) under nitrogen atmosphere at room temperature with 390 nm LEDs irradiation for 24 h. Yields were determined by [1]H NMR analysis of crude products using 1,3,5-trimethoxybenzene as internal standard.

-OMe (**1u**), -COOMe (**1v**), and -CH₂OH (**1w**), et. al, at the benzylic position could be tolerated well, resulting corresponding products in up to excellent yields. Notably, the tolerated vinyl (**1t**), carbonyl, and hydroxyl groups (**1w**) could be used for the synthesis high-value compounds. In addition, when a substrate bearing both methyl and ethyl groups (**1y**) was explored, alkylation products could be obtained with a ratio of 1:1.2. To our delight, this EDA complex catalysis would be further used for the alkylation of complex substrates, such as *Desoxyanisoin, Dapaglifizoin* and *Epiandrosterone* deriviation (**1z-ab**). Regretfully, a substrate with a tertiary C(sp³)−H bond (**1ac**) failed to afford the desired product, probably due to the steric hindrance.

Moreover, the scope of electron-deficient alkenes was explored (Fig. 5). A collection of benzylidenemalononitriles bearing electron-donating substituents (**2b-2e**) and electron-withdrawing substituents (**2f-2i**) at the *para*-position of the aromatic ring could be employed in the alkylation with 4-methylanisole (**1a**) in moderate to good yields (28−79%). A substrate with a strong electron-donating ethoxyl group (**2d**) was less efficient, probably due to that the electron-rich ethoxylbenzene ring being interacted with the electron-poor catalyst competingly against the substrate **1a**. Gladly, the useful but reactive functional groups including Br and Bpin groups could be remained during the transformations, providing great potentials for subsequent transition metal-catalyzed couplings. A substrate bearing an ester group in the *meta*-position of benzene ring (**2k**) was also proceeding smoothly with **1a** to afford the corresponding product in 70% yield. Besides phenyl substituents, other aryl groups, such as 2-naphthyl (**2m**) and 2-thienyl groups (**2n**), were suitable for the transformation with 2,6-dimethoxytoluene, obtaining the products in moderate yields. The reaction of **1a** with **2j**, **2m** and **2n** afforded lower yields, probably due to the competing interaction of **2j**, **2m** and **2n** with 4-nitrophthalonitrile prevented from the formation of EDA complex of **1a** and **cat1**. Interestingly, the substrate containing a tetraphenylethylene moiety, which was well-known as a powerful aggregation-induced emission (AIE) agent[26], could also lead to the desired product, albeit in a relatively low yield. Moreover, the electron-deficient groups could be expanded to sulfonyl (**2p**) and ester groups (**2q**). Regretfully, other less electron-deficient alkenes, such as diethyl

2-benzylidenemalonate, diethyl vinylphosphonate, rylonitrile, (vinylsulfonyl)benzene, methyl acrylate, cyclohex-2-en-1-one, cyclopent-2-en-1-one, and furan-2,5-dione could not react with 4-methylanisole.

This benzylic C−H cleavage strategy via EDA complex catalysis was also found to be efficient for the oxidation of benzylic alcohols under the aerobic condition in DMSO (see more details in Supplementary Table 2, Supplementary Information)[27–30]. The trapping of the radical intermediate **INT A** (from the catalytic cycle) with DMSO or O₂ would be responsible for the oxidation. An array of electron-rich benzyl alcohols could be oxidized to aldehydes smoothly (Fig. 6). All the *para*-, *ortho*-, and *meta*-anisyl alcohols could be converted into desired aldehydes, while the *meta*-anisyl alcohol observed the lowest efficiency. These results indicated that the electron density distribution have tremendous impact on the efficiency of single electron transfer (SET). The more e-rich substrate could obtain the higher oxidation efficiency, since the reaction of 3,4-dimethoxybenzyl alcohol (**5d**) can achieve the quantitative yield. The electron-rich benzyl alcohol bearing functional groups, such as -Br (**5e**), -Cl (**5 f**), -SMe (**5 g**) and -OPh (**5 h**) could undergo oxidation smoothly (74-94% yields). Other aryl groups, such as 2-nathyl (**5i**) and benzo[*b*]thiophenyl (**5j**) could be tolerated under optimized condition and furnished aldehyde with 50-76% yield. Furthermore, a various of secondary benzyl alcohols were employed to give the corresponding ketones (**5k-n**) including the hCECE inhibitor 4,4'-Dimethoxybenzil (**5n**).

The hydrocarboxylation of alkenes using formate salts has recently drawn chemists' significant attentions[31–43]. The successful catalytic e-rich aromatics activation via π-π stacking had driven us to examine other reactive e-rich partner, such as formate salts. Gladly, 4-nitrophthalonitrile was found to be interacted with potassiom formate via π-anion stacking to generate corresponding electron donor-acceptor complex in ground state, which could be confirmed by both UV-Vis spectra and [1]H NMR analysis (see Supplementary Figs. 17-18 in SI). After being excited under 390 nm LED, the single electron transfer (SET) and self-deprotonation happened subsequently to give carbon dioxide radical anion (CO₂•⁻), which could act as carboxylative source by addition to alkene. This radical addtion process could be captured by TEMPO, which could be detected by HRMS (see Supplementary Fig. 16

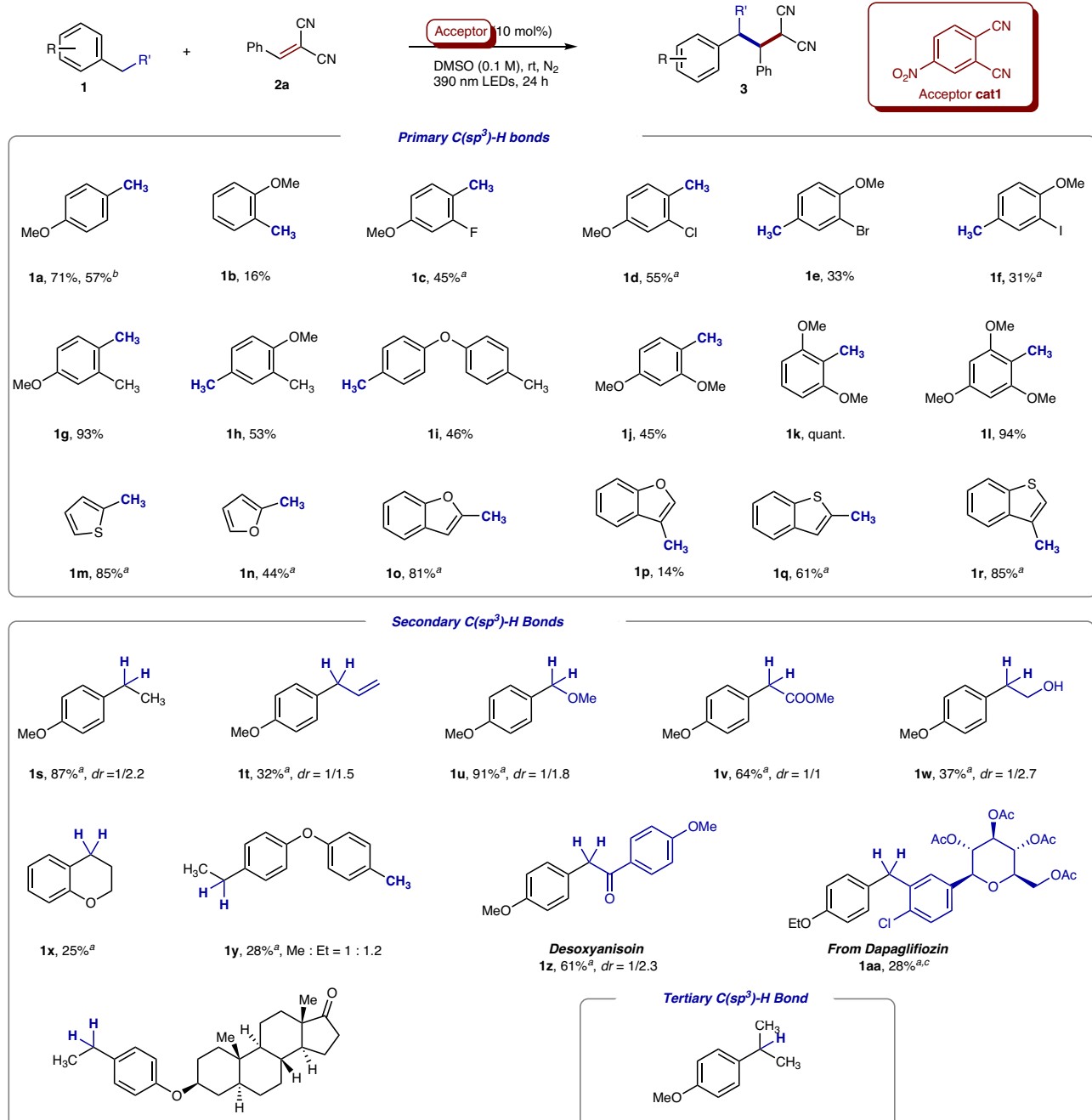

**Fig. 4 | Scope of electron-rich arenes.** Reactions were conducted with **1** (0.4 mmol), **2a** (0.2 mmol), 4-nitrophthalonitrile **cat1** (0.02 mmol) in DMSO (2 mL) under nitrogen atmosphere at room temperature with 390 nm LEDs irradiation for 24 h. All yields were calculated after isolation. [a]The reaction time was prolonged to 72 h. [b]**2a** (20 mmol) was used. [c]Four diastereomers were observed but the ratio was difficult to determine.

in SI). A series of carboxylic acid could be obtained using this π-anion stacking catalysis (Fig. 7). Diverse styrenes bearing electron-donating and electron-withdrawing substituents furnished the anti-Markovnikov hydrocarboxylation reaction in moderate to excellent yields[31–43]. Both 1,1- and 1,2-disubstituted ethylene derivatives could be used. Styrenes bearing halogen atom (**8d**), methoxy (**8e**), and phenyl groups (**8 f**) in the *para*-position and methyl (**8 g**) in the *ortho*-position were tolerated well to give corresponding acids. β-Methyl styrene (**8i, 8j**) and β-phenyl styrene (**8k**) were employed under optimized conditions to give desired acids selectively. Interestingly, when phenylacetylenes were used under standard conditions, the hydrocarboxylation accompanied with

hydrogenation happened to give phenylpropionic acid in 55% yield (palsusible mechanism see Supplementary Fig. 19 in SI). Other terminal and internal aryl-substituted alkynes were then examined to obtained corresponding carbonyxlic acids.

## Mechanistic studies and plausible mechanism
A series of additional mechanistic studies were then performed to gain comprehensive insight into the mechanism of the alkylation reaction (Fig. 8)[44–46]. The control experiment in the absence of light demonstrated that light should play an indispensable role in the transformation (Fig. 8a and Supplementary Table 1, entry 3), which could further be

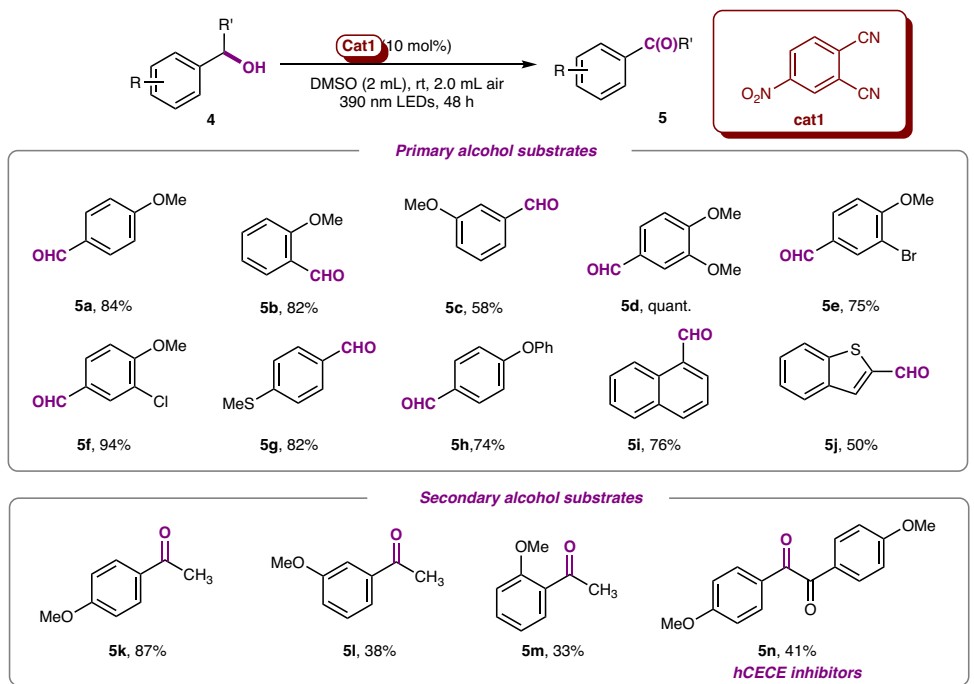

**Fig. 5 | Scope of alkenes.** Reactions were conducted with **1a** (0.4 mmol), **2a** (0.2 mmol), 4-nitrophthalonitrile **cat1** (0.02 mmol) in DMSO (2 mL) under nitrogen atmosphere at room temperature with 390 nm LEDs irradiation for 24 h. All yields were calculated after isolation. [a]Reactions were conducted with **1k** (0.4 mmol) and **2** (0.2 mmol).

**Fig. 6 | Scope of the electron-rich benzyl alcohols oxidation.** Reactions were conducted with **4** (0.2 mmol), 4-nitrophthalonitrile **cat1** (0.02 mmol) in DMSO (2 mL) under 2.0 mL air atmosphere at room temperature with 390 nm LEDs irradiation for 48 h. All yields were calculated after isolation.

confirmed by the light on-off experiments (Supplementary Table 5). The overall quantum yield (Φ) was determined as 0.384 (<1.0), indicating that a radical chain propagation is unlikely to be involved in the transformation. The UV-Vis spectra of 4-methylanisole (**1a**), benzylidenemalononitrile (**2a**), 4-nitrophthalonitrile (**cat1**), **1a/2a**, and **1a/cat1** in same concentrations with the reaction mixture were then examined (Fig. 8b). The corresponding UV-Vis absorption showcased the increasing absorption at the λ = 350-500 nm, indicating the formed EDA complex

possessed the redshiffed visible-light absorptions to facilitate the photo-excitation. The solution of **1a** and **cat1** became pale-yellow, further confirming the interaction between each other. Notably, the mixture of **1a** and **2a** also presented a relative smaller increasing absorption compared with **1a/cat1**, which could explain why the reaction in the absence of **cat1** could proceed in lower efficiency.

On the other hand, the radical trapping studies of **1a** in the presence of TEMPO were conducted under the standard conditions. The TEMPO-

**Fig. 7 | Scope of the hydrocarboxylation.** Reactions were conducted with **6** or **7** (0.2 mmol), potassium formate (2.0 mmol), 4-nitrophthalonitrile (0.01 mmol) in DMSO (2 mL) under nitrogen atmosphere at room temperature with 390 nm LED irradiation for 72 h. Reaction mixtures were acidified before isolation. All yields were calculated after isolation.

adduct product **3a-TEMPO** was observed by HRMS spectra either in the presence or absence of compound **2a** (Fig. 8c), indicating a radical process was involved for the formation of a benzylic radical during the transformation. The reaction in the presence the radical scavenger butylated hydroxytoluene (BHT) observed the significant yield decreasing, further confirming the radical process (Fig. 8d). Moreover, the intermolecular kinetic isotope effect experiment was performed by treating excess **1a** and **1a-D** with **2a** under the standard conditions. The KIE was determined to be $k_H/k_D = 5.5/1$, suggesting that the C − H bond cleavage might be the turnover-determining step (Fig. 8e).

Furthermore, the cyclic voltammetry studies were explored to afford the redox potential of 4-nitrophthalonitrile and **1a** (see SI), indicating that 4-nitrophthalonitrile could act as a proper redox catalyst (catalytic electron acceptor) ($E_{1/2} = −1.04$ (*vs*. Fc$^+$/Fc)) while **1a** was unstable during the CV test ($E_{p/2} = 1.19$ (*vs*. Fc$^+$/Fc)) probably due to the potential benzylic C − H cleavage.

On the basic of experimental results above, we proposed a photoexcitation electron-donor-acceptor (EDA) complex mechanism involving acceptor catalysts depicted in Fig. 9. The initiated step was the π−π interaction between acceptor catalysts **cat1** and 4-methylanisole to form an EDA complex. Upon irradiation under 390 nm LEDs, the corresponding excited state triggered single electron transfer (SET) would lead to catalyst radical anion (**cat-A**) and 4-methylanisole radical cation (**1a-A**). The later one then underwent facile deprotonation, resulting in the formation of the benzylic radical intermediate (**Int A**). The subsequent radical addition of **Int A** with the electron-deficient alkene (**2a**) afforded the alkyl radical (**3aa-A**), which interacted with the catalyst radical anion (**cat-A**) to form alkyl anion **3aa-B** and turn over the acceptor catalyst (pathway I). The simultaneously protonation also furnished the final coupling product **3aa**.

Since the quantum yield was less than 1.0, another potential mechanism involving the hydrogen atom transfer (HAT) between **3aa-A** with **1a** (pathway II) was less possible.

## Synthetic Applications

Finally, the synthetic utility of this strategy was then explored (Fig. 10)[47]. Firstly, the 1*H*-pyrazole-3,5-diamine derivative **9** could be prepared in quantitative yield by the cyclization reaction of **3aa** with hydrazine hydrate under the reflux conditions. Moreover, the alkylation product **3aa** could be further transformed to acid **10** and ester **11** in moderate yields by the oxidation of the malonitrile groups with DMSO/Air or *m*-CPBA. In addition, the aerobic oxidation of **3aa** in the presence of 1-methylpiperazine produced the amide **12**, which is known as a M$_1$ antagonist[48].

## Discussion

In sum, we have characterized a cocrystal of 4-nitrophthalonitrile and 1,3,5-trimethoxybenzene. This electron-donor-acceptor (EDA) complex emerged the strong visible light absorption at the $\lambda = 389 - 457$ nm and energy profiles on the electron-hole distribution of the EDA aggregate solidly supported the readily electron transfer between the donor and acceptor. Utilizing 4-nitrophthalonitrile as a catalytic acceptor, the benzylic C−H bond cleavage could be achieved by photoactivation. The Giese reaction of alkylanisoles and the oxidation reaction of the benzyl alcohols were then developed efficiently with a wide scope of substrates. Moreover, this e-poor catalyst could further activate formate salts smoothly and furnish the hydrocarboxylation reaction of alkenes and alkynes using potassium formate as a carbonxylation source. A mechanism involving the catalytic photoactivation of an EDA complex and deprotonation was proposed based

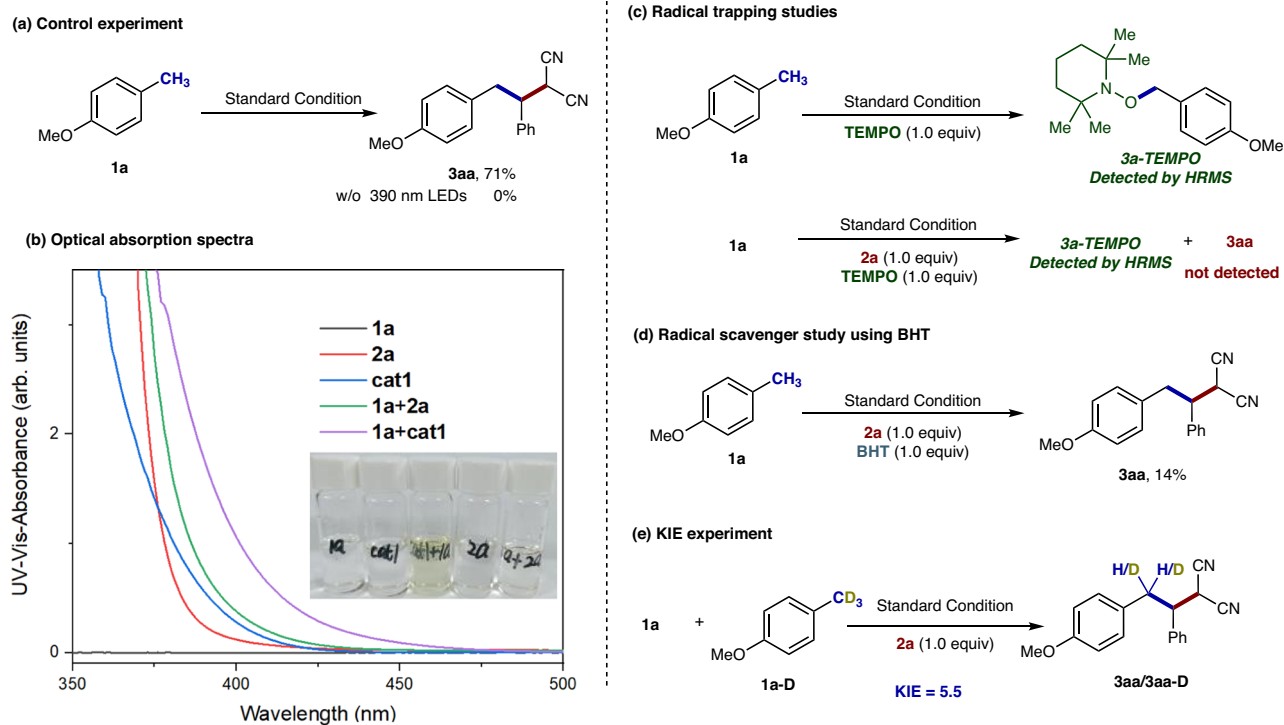

**Fig. 8 | Mechanistic experiments for the alkylation. a** The light-free experiment. **b** The optical absorption spectra. **c** The radical trapping studies. **d** The radical scavenger studies using BHT. **e** The KIE experiment.

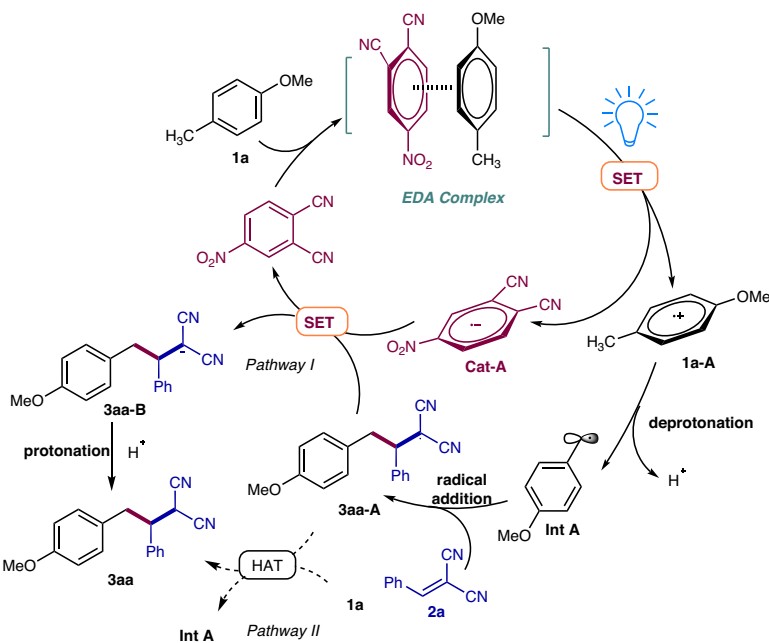

**Fig. 9 | Plausible mechanism.** The catalytic cycle includes EDA complex formation, photoinduced single electron transfer (SET), deprotonation, radical addition, and protonation.

on the mechanistic studies. Further studies and application of the catalytic electron acceptor are ongoing in our lab.

## Methods

### General procedure for synthesis of product 3

To a 4 mL vial were added **1** (0.4 mmol), **2** (0.2 mmol), and 4-nitrophthalonitrile (0.02 mmol) in DMSO (2.0 mL) in an $N_2$ glovebox. The vial was sealed and transferred out of glovebox. Under irradiation

at 390 nm LEDs, the resulting mixture was stirred for 24-72 hours at rt. Saturated brine was then added and the aqueous layer was extracted with ethyl acetate (10 mL × 3). Evaporation and flash chromatography on silica gel afforded **3**.

### General procedure for synthesis of product 5

To a 4 mL vial were added **4** (0.2 mmol), and 4-nitrophthalonitrile (0.02 mmol) in DMSO (2.0 mL) in air. Under irradiation at 390 nm

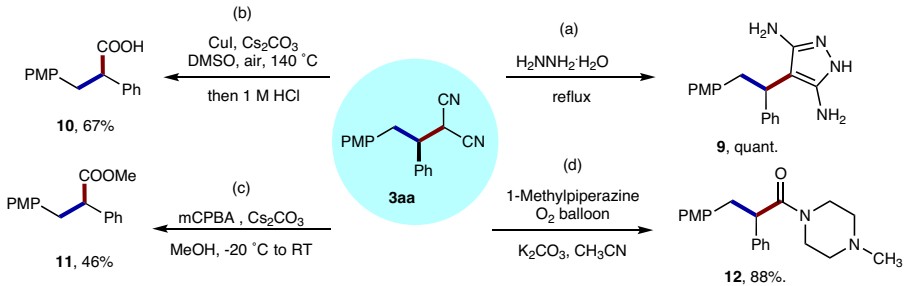

**Fig. 10 | Synthetic application of compound 3aa. a** The cyclization of **3aa** with hydrazine hydrate. **b** The oxidation of **3aa** to acid **10**. **c** The oxidation of **3aa** to ester **11**. **d** The oxidative amidation of **3aa** to amide **12**.

LEDs, the resulting mixture in 2.0 ml air was stirred for 48 hours at rt. Saturated brine was then added and the aqueous layer was extracted with ethyl acetate (10 mL × 3), Evaporation and flash chromatography on silica gel afforded **5**.

### General procedure of 8
To a 4 mL vial were added **6** or **7** (0.2 mmol), HCOOK (2.0 mmol), and 4-nitrophthalonitrile (0.01 mmol) in DMSO (2.0 mL) in an $N_2$ glovebox. The vial was sealed and transferred out of glovebox. Under irradiation at 390 nm LEDs, the resulting mixture was stirred for 72 hours at rt. The mixture was acidified with 2 mL dilute HCl (2 N) and quenched with $H_2O$ and the aqueous layer was extracted with ethyl acetate (10 mL × 3). Evaporation and flash chromatography on silica gel afforded **8**.

### Data availability

The X-ray crystallographic coordinates for structures reported in this study have been deposited at the Cambridge Crystallographic Data Centre (CCDC), under deposition numbers 2277272. These data can be obtained free of charge from The Cambridge Crystallographic Data Centre via www.ccdc.cam.ac.uk/data_request/cif. All the data supporting the findings of this study are available within the article and its Supplementary Information, or from the corresponding author on request. Source data of the coordinates of the calculated optimized structures are present. Source data are provided with this paper.

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

## Acknowledgements

R.Z. is a Xiaomi Young Scholar and is grateful for the financial support from the National Natural Science Foundation of China (22371223) and the startup funds from Xi'an Jiaotong University (XJTU). S.F.N. acknowledge funding from the STU Scientific Research Foundation for Talents (NTF20022). We thank Ms. Shuang Wei in this group for reproducing the results of **3ga** and **5i**. We also thank Dr. Xiaolong Yang from XJTU for the discussion on electron donor-acceptor luminescence.

## Author contributions

R.Z. supervised the project. R.Z. and T.X. designed the experiments. T.X. performed and analyzed the experiments. C.X. analyzed the CV curves. C.M. and S.-F.N. performed the DFT calculation. T.X., L.L., C.X., S.-F.N., and R.Z. prepared this manuscript.

## Competing interests

The authors declare no competing interests.
