## [Peer Review File · Nature Communications]

Characterization of A π - π Stacking Cocrystal of 4-Nitrophthalonitrile Directed Toward Application in PhotocatalysisReviewers' Comments:

Reviewer #1:

Remarks to the Author:

Organic cocrystal strategy is an effective approach to achieve radicals through photoexcitation, but it has not received enough attention in the field of photocatalysis. In this work, EDA complex is highly favorable to charge transfer under light irradiation. The EDA catalyst could be efficiently applied in the benzylic C-H bond photoactivation. In general, I recommend its publication after major revisions as noted below:

Comment 1: Regarding Figure 1b, if you need to compare the absorption range of materials, you can choose quantitative analysis or normalization to make the data more readable.

Comment 2: The authors further demonstrated the radical process through free radical trapping experiments, meanwhile EPR testing is also necessary.

Comment 3: As for Figure 4, the signal accumulation of carbon radicals in EPR is not clear enough, so can you provide more intuitive evidence?

Comment 4: In photocatalytic experiments, what role does 1,3,5-trimethoxybenzene play in the reaction besides being an electron donor? How to exclude the influence of donor molecule on the reaction.

Reviewer #2:

Remarks to the Author:

Review attached.

The manuscript by Ni and Feng describes the use of 4-nitrophthalonitrile as a catalytic acceptor in EDA complex-mediated photoinduced reactions. Whilst there are many examples of EDA complex-mediated photocatalysis with catalytic donors, there are comparatively few examples of catalytic acceptors, which adds to the novelty of this work. The authors begin with the isolation and characterization of a co-crystal of 4-nitrophthalonitrile and trimethoxybenzene, which they then use as inspiration for designing various reactions of electron-donor molecules in photoinduced reactions catalyzed by 4-nitrophthalonitrile.

The reactions include benzylic C-H alkylations of electron-rich alkylbenzenes with benzylidenemalononitriles, aerobic oxidation of electron-rich benzylic alcohols to aldehydes, hydrocarboxylations of styrenes, and reductive hydrocarboxylations of aryl acetylenes. Apart from the reductive hydrocarboxylations of aryl acetylenes, these transformations have already been reported under similar conditions with alternative photoredox catalysts (alkylation: reference 28; oxidation: *ACS Catal.* **2018**, *8*, 5425; hydrocarboxylation: reference 38). As a result, demonstration of the novelty of the work relies on good experimental support for the proposed mode of catalysis wherein 4-nitrophthalonitrile acts as a catalytic acceptor in EDA complex-mediated reactions, rather than a photoredox catalyst. However, I do not think adequate support for this mechanism has been given. Firstly, for the C-H alkylation reaction, the UV/Vis studies in Figure 3b require 1000 equivalents of trimethoxybenzene for a small bathochromic shift to occur, which is not representative of the reaction conditions used. Secondly, the wavelength of light used in all the transformations is 390 nm, which is capable of directly exciting 4-nitrophthalonitrile (according to the UV/Vis spectra in Figure 3b). Therefore, it is more likely that 4-nitrophthalonitrile is simply acting as a photocatalyst, either reacting through outer-sphere SET or HAT (see *J. Am. Chem. Soc.* **2023**, *145*, 2794 and *Org. Lett.* **2023**, *25*, 6517 for the HAT mechanism, which is supported by the importance of the nitro group when comparing **cat2** to **cat3** in Table 1), which significantly detracts from its novelty.

In its current form, I do not think this work achieves the scientific advance required for publication in Nature Communications, therefore, I do not support publication. The authors should consider the additional comments below before resubmitting to another journal.

Additional comments/queries:

- 1) The title is somewhat misleading because the co-crystal is not used in the photocatalysis.
- 2) Lines 51-52: "These reactions all suffer from the less variety of acceptor catalysts" – this is not clear. Why is the variety of catalysts important?
- 3) Lines 58-60: "Firstly, the cyano group of the dicyanobenzene derivatives are known to be altered in excited states, which could change their electronic properties." – this is not clear. How is the cyano group altered? Why is changing the electronic properties advantageous?
- 4) Lines 60-61: "In addition, the planar structure is less hindered in the vertical axis, offering great opportunity for the π - π interaction" – this is also not clear. No other acceptors have been discussed in this paragraph, so what is the planar structure less hindered than?
- 5) Figure 1b: Isolation of a crystalline EDA complex with trimethoxybenzene does not necessarily mean that a similar complex is formed in DMSO. As a result, comparison of the UV/Vis of the solid-state co-crystal with solutions of 4-nitrophthalonitrile and trimethoxybenzene in DMSO is misleading. For accurate comparison, the co-crystal should be dissolved in DMSO (0.01 mM). Similarly, the solid-state EPR does not necessarily translate to solution-phase photochemical reactions.
- 6) Table 1: The reaction also gives a significant amount of product in the absence of catalyst (27%). This result should be discussed, since it means that **cat3-cat5** are not promoting the reaction.
- 7) Table 1: The structure of the Eosin Y disodium salt is incorrect. Also, the neutral eosin Y should not be depicted as a spirocycle because this is not the photoactive isomer.
- 8) Product **3pa** is missing from Table 2.
- 9) Lines 146-148: "when the substrate bearing both methyl and ethyl groups (**1y**) was explored, the alkylation products could be obtained with a ratio of 1:1.2, indicating that the secondary C(sp³)-H bond functionalization thermodynamic favorable." – A ratio of 1:1.2 means the reaction is unselective, therefore this discussion of which regioisomer is thermodynamically favored should be removed.

- 10) Lines 194-195: "In addition, the aerobic oxidation of **3aa** in the presence of 1-methylpiperazine produced the amide **7**, which is known as a M1 antagonist" – a reference is needed for this.
- 11) I do not understand the UV/vis data in Figure 3b. The donor is not one of the substrates used in the alkylation chemistry and there are no details about the concentrations or solvent used. Also, the acceptor/donor ratios look wrong. The ratio of acceptor to donor under the optimised reaction conditions is 20:1, so why is a ratio of 1000:1 used for the UV/Vis studies? My conclusion from the UV/vis data is that no significant EDA complex formation occurs and that the acceptor **cat1** is directly excited by 390 nm light.
- 12) What is the mechanism for the reductive hydrocarboxylation of alkynes? It would be useful to include a proposed mechanism in the SI.

Supporting information:

- 13) Check all figure and table numbers in the SI. There are repeat numbers (e.g., 4 x Figure S4) and Table 5 should be re-numbered "Table S3" to match the format of the other tables.
- 14) Figure S9: It is not clear why the different ratios of **cat1**:HCOOK have been used. It looks like a Job plot is being constructed but no discussion is given.
- 15) Figure S9: The solvent used for these studies should be stated.
- 16) Figure S10: The concentrations for these NMR studies should be included. They should ideally match the reaction conditions.
- 17) Figure S10: Check the ratio of **cat1**:HCOOK in the second spectrum. Based on the trends, the ratio should be reversed.
- 18) Figure S10: I would not expect such a dramatic change in the chemical shifts for the protons in nitrophthalonitrile upon formation of an EDA complex. The broadening of the signals between 8.4-9 ppm suggests an interaction, however, the well-resolved new signals from 6.4-7.3 ppm could result from a reaction. The dramatic chemical shift change looks like the nitro group could have been converted to an alcohol, giving 3,4-dicyanophenoxide under the basic conditions. Re-isolation of nitrophthalonitrile is required to confirm that no reaction is occurring.
- 19) P. S73: Product **3pa** does not look clean. There are 6 aliphatic signals when there should only be 4.
- 20) P. S80-S86: The compound numbers for the structures on these spectra are wrong.
- 21) P. S81: The structure of the spectrum of **3xa** state 1:1.52 dr but it is not clear from the NMR how this was determined (it looks like a single diastereomer). The dr for this product should also be added to the scheme on P. S20 and to Table 2 in the manuscript.
- 22) The dr's are also missing for products **3aaa** and **3aba** in both the SI and manuscript.
- 23) P. S104-S107: Check the numbers for the structures on the spectra.

Reviewer #3:

Remarks to the Author:

In this report Xue et al. present a novel electron acceptor molecule that can participate in photocatalytic processes with suitable electron rich donor substrates. The activation proceeds via the generation of an electron-donor-acceptor (EDA) complex, which can absorb light acting as photocatalyst for various reactions. EDA complexes have been extensively used in photochemistry, but mostly as transient photoactive species generated between reactants or transient intermediates. The use of EDA acceptors as catalysts is somehow limited. In this regard, the use of highly acidic boron species was studied in the past (as correctly mentioned by the authors), and little is known about electron poor aromatic species, with systematic studies performed only by Melchiorre and others (see ref 21) on specific redox active esters. Although presenting some clear scope limitations (a methoxy group is needed in the acceptor aromatic ring and often highly activated reagents are used, e.g., dicyano Michael acceptors) the novel EDA acceptor photocatalyst presented by the authors presents a clear conceptual interest. In addition, the characterization study, including an EDA crystal structure, is a clear advance in the area.

Therefore, I recommend publication of this report in Nature Communications, pending the following revisions/comments below:

- The narrative of this paper seems "fragmented", presumably due to the choice of presenting first the Giese transformation and mechanism, followed by other transformations which seem "isolated" and poorly described. Perhaps simply rearranging the structure presenting all the transformations first (e.g., the potential of this photocatalyst), followed by a mechanistic study at the end of the paper using the Giese reaction as benchmark process for mechanistic study, would improve the reading.
- Indoles are usually good in establishing EDA complexes. Would Me-indole work as a substrate for this reaction?
- The following sentence is slightly confusing and should be reworded: "The overall quantum yield (Φ) was determined as 0.384 (<1.0), indicating that the radical chain growth pathway was also lowly possibly involved during the transformation." In "indicating that a radical chain propagation is unlikely to be involved in the transformation...."
- Table 3: 1d is incorrectly numbered – this should be 1k
- Table 3 and line 162: numbering incorrect – electron deficient substrates are 2f-2i but authors write in text 2f-2g.
- Line 172-173 and Table 3: the authors should provide an explanation for why 1a is not suitable with substrates 2m and 2n.

Recommendations for the supporting information:

- The authors claim that the reaction is performed at room temperature, however the picture for the reaction set up shows no fan is used or any cooling mechanism to maintain room temperature while 2 lamps are in operation. The temperature should be measured and presented to facilitate reproducibility.
- Compounds 3ma, 3pa and 3ua are contaminated with impurities and should be repurified, and spectra re-recorded in a revised version.
- The NMR spectrum of compound 3va is not properly integrated
- Finally, ensure the numbering is consistent and correct. Many compounds are numbered incorrectly or have a different number assigned to the NMR spectra.

Re: Reviewer 1:

Original comment: Organic cocrystal strategy is an effective approach to achieve radicals through photoexcitation, but it has not received enough attention in the field of photocatalysis. In this work, EDA complex is highly favorable to charge transfer under light irradiation. The EDA catalyst could be efficiently applied in the benzylic C-H bond photoactivation. In general, I recommend its publication after major revisions as noted below:

Response: We firstly would like to thank reviewer 1's recognition and acknowledge reviewer 1's kind efforts in analyzing our manuscript and providing all these evaluable and constructive suggestions in great details.

Original comment 1: Regarding Figure 1b, if you need to compare the absorption range of materials, you can choose quantitative analysis or normalization to make the data more readable.

Response: We thank this constructive suggestion and have normalized the UV-Vis-absorption spectra to make the data more readable.

Original comment 2: The authors further demonstrated the radical process through free radical trapping experiments, meanwhile EPR testing is also necessary.

Response: Previously, we have conducted the radical trapping experiments using TEMPO and indeed observed the TEMPO-adduct by HRMS. These results had been presented in Figure 3C.

(c) Radical trapping studies

On the other hand, following the suggestion, the electron paramagnetic resonance (EPR) spectroscopy was employed to directly detect radical intermediates. Firstly, using the cocrystal of 4-nitrophthalonitrile and 1,3,5-trimethoxybenzene as starting sample, no radical signal was recorded without irradiation. Interestingly, once the sample was irradiated with 390 nm light (ex situ of the magnet), a build-up of a radical signal, assigned as a carbon radical ($g = 2.003$), was observed in the EPR spectrum. When the saturated solution of 4-nitrophthalonitrile and 1,3,5-trimethoxybenzene in DMSO was examined under irradiation, a similar signal could be obtained, however, in a relatively low resolution, probably due to the low concentration of radical species. Delightedly, a much higher resolution signal could be obtained by switching the solvent to DMF. These results indicated that the electron transfer might indeed proceed between 4-nitrophthalonitrile and 1,3,5-trimethoxybenzene to general the carbon radical species under the irradiation of 390 nm light.

Original comment 3: As for Figure 4, the signal accumulation of carbon radicals in EPR is not clear enough, so can you provide more intuitive evidence?

Response: We have retested the EPR spectrum of the cocrystal, and a much clearer signal have been collected.

Original comment 4: In photocatalytic experiments, what role does 1,3,5-trimethoxybenzene play in the reaction besides being an electron donor? How to exclude the influence of donor molecule on the reaction.

Response: 1,3,5-Trimethoxybenzene did not play any role in the catalytic reactions. It was only a model substrate for forming a π - π stacking cocrystal with 4-nitrophthalonitrile, and we did not add any 1,3,5-trimethoxybenzene during the catalytic transformations.

Re: Reviewer 2

Original comment: The manuscript by Ni and Zeng describes the use of 4-nitrophthalonitrile as a catalytic acceptor in EDA complex-mediated photoinduced reactions. Whilst there are many examples of EDA complex-mediated photocatalysis with catalytic donors, there are comparatively few examples of catalytic acceptors, which adds to the novelty of this work. The authors begin with the isolation and characterization of a cocrystal of 4-nitrophthalonitrile and trimethoxybenzene, which they then use as inspiration for designing various reactions of electron-donor molecules in photoinduced reactions catalyzed by 4-nitrophthalonitrile.

Response: We firstly appreciate reviewer 2's time and efforts on examining this manuscript in details.

Original comment: The reactions include benzylic C-H alkylations of electron-rich alkylbenzenes with benzylidenemalononitriles, aerobic oxidation of electron-rich benzylic alcohols to aldehydes, hydrocarboxylations of styrenes, and reductive hydrocarboxylations of aryl acetylenes. Apart from the reductive hydrocarboxylations of aryl acetylenes, these transformations have already been reported under similar conditions with alternative photoredox catalysts (alkylation: reference 28; oxidation: *ACS Catal.* **2018**, *8*, 5425; hydrocarboxylation: reference 38). As a result, demonstration of the novelty of the work relies on good experimental support for the proposed mode of catalysis wherein 4-nitrophthalonitrile acts as a catalytic acceptor in EDA complex-mediated reactions, rather than a photoredox catalyst.

Response: We firstly would like to thank reviewer 1's recognition that 4-nitrophthalonitrile acts as a catalytic acceptor in EDA complex reactions rather than a photoredox catalyst. The alkylation, aerobic oxidation, and hydrocarboxylation reactions were indeed reported using other catalytic systems through photoredox catalysts, however, there is no reports on such an EDA complex catalysis using 4-nitrophthalonitrile as a catalytic acceptor, which brought significant novelty.

Original comment: However, I do not think adequate support for this mechanism has been given. Firstly, for the C-H alkylation reaction, the UV/Vis studies in Figure 3b require 1000 equivalents of trimethoxybenzene for a small bathochromic shift to occur, which is not representative of the reaction conditions used.

Response: Nice concern. First, the reason we used 1000 equivalents of trimethoxybenzene is just because we would like to show the significant changes in spectra. Higher equivalents of trimethoxybenzene could drive the chemical equilibrium to form higher concentration of EDA complex in solution, which would cause more distinguishable red shift in UV/Vis spectra.

Second, to address such a concern, we have also re-examined UV-Vis spectra of 4-methylanisole (**1a**), benzylidenemalononitrile (**2a**), 4-nitrophthalonitrile (**cat1**), **1a/2a**, and **1a/cat1** with same concentrations of the reaction mixture. The corresponding UV-Vis absorption of **1a/cat1** showcased the increasing absorption at the $\lambda = 350 \sim 500$ nm, indicating the formed EDA complex possessed the new visible-light absorptions to facilitate the photo-excitation. The solution of **1a** and **cat1** became pale-yellow, further confirming the interaction between each other. Notably, the mixture of **1a** and **2a**

also presented a relative smaller increasing absorption compared with **1a**/**cat1**, which could explain why the reaction in the absence of **cat1** could proceed in lower efficiency.

These new results have been added in the Figure 3b and discussed in main text.

Original comment: Secondly, the wavelength of light used in all the transformations is 390 nm, which is capable of directly exciting 4-nitrophthalonitrile (according to the UV/Vis spectra in Figure 3b). Therefore, it is more likely that 4-nitrophthalonitrile is simply acting as a photocatalyst, either reacting through outer-sphere SET or HAT (see *J. Am. Chem. Soc.* **2023**, 145, 2794 and *Org. Lett.* **2023**, 25, 6517 for the HAT mechanism, which is supported by the importance of the nitro group when comparing **cat2** to **cat3** in Table 1), which significantly detracts from its novelty.

Response: We really appreciate reviewer 2's concern on the mechanism. However, we cannot fully agree with it, since the current experiments support very well the EDA complex catalysis instead of "outer-sphere SET or HAT". Here are reasons:

(1) The reaction was quite sensitive to the electron density of used C-H substrates, since electron-rich aromatics are typically required. We have examined the reactions of toluene or cyclohexane with **2a**, however, no desired alkylation products could be obtained at all. These results are very compatible with π - π stacking EDA complex but conflict with the HAT mechanism, since the later one was not affected significantly by the electron density of the substrates.

(2) The steric effect did not play a crucial role in reaction efficiency. For example, the steric-hindrance substrates 1,3-dimethoxy-2-methylbenzene (**1k**) could undergo alkylation with **2a** in a higher yield (quant.) than 4-methylanisole **1a**, indicating that electron density instead of steric hindrance played a crucial role. These results are very compatible with π - π stacking EDA complex but conflict with the HAT mechanism.

(3) The UV-Vis spectra are much matching with EDA complex catalysis. We have also re-examined UV-Vis spectra of 4-methylanisole (**1a**), benzylidenemalononitrile (**2a**), 4-nitrophthalonitrile (**cat1**), **1a/2a**, and **1a/cat1** with same concentrations of the reaction mixture. The corresponding UV-Vis absorption of **1a/cat1** showcased the increasing absorption at the $\lambda = 350 \sim 500$ nm, indicating the formed EDA complex possessed the new visible-light absorptions to facilitate the photo-excitation. The solution of **1a** and **cat1** became pale-yellow, further confirming the interaction between each other.

(4) The EPR spectra are much matching with EDA complex catalysis since only carbon radical was observed during EPR examination. Firstly, using the cocrystal of 4-nitrophthalonitrile and 1,3,5-trimethoxybenzene as starting sample, no radical signal was recorded without irradiation. Interestingly, once the sample was irradiated with 390 nm light (ex situ of the magnet), a build-up of a radical signal, **assigned as a carbon radical ($g = 2.003$)**, was observed in the EPR spectrum. When the saturated solution of 4-nitrophthalonitrile and 1,3,5-trimethoxybenzene in DMSO was examined under irradiation, a similar signal could be obtained, however, in a relatively low resolution, probably due to the low concentration of radical species. Delightedly, a much higher resolution signal could be obtained by switching the solvent to DMF. These results indicated that the electron transfer might indeed proceed between 4-nitrophthalonitrile and 1,3,5-trimethoxybenzene to general the carbon radical species under the irradiation of 390 nm light.

(5) The **cat1** could be used catalytically. In the literature, stoichiometric amount of nitroarenes have been used for the reported photocatalyzed oxidative transformation (Nature 2022, 610, 81-86; J. Am. Chem. Soc. 2023, 145, 2794; Org. Lett. 2023, 25, 6517). The nitroarenes would consume during the reaction with the side product nitrosoarene generated. However, in our catalytic EDA complex photoactivation strategy, only 10 mol% of 4-nitrophthalonitrile was needed, and the catalysis would be maintained after the transformation.

(6) Most importantly, the reaction of **1a** with benzylidenemalononitrile **2a** might actually proceed under irradiation of 460 nm light. According to UV-Vis spectra, 4-nitrophthalonitrile (**cat1**) did not absorb 460 nm light, while the EDA complex represent small absorption. After switching the light source to 460 nm LEDs, the C-H alkylation product could be obtained in 20% yield successfully. Without **cat1** under irradiation of 460 nm light, only trace amount of **3aa** could be obtained. These results are much more matching with EDA complex catalysis.

Based on all these reasons, we insist in the mechanism involving EDA complex catalysis. And all these results have been presented in the proper part in main text or supplementary information.

Additional comments/queries:

1) The title is somewhat misleading because the cocrystal is not used in the photocatalysis.

Response: Thank you for the suggestion, we have revised the title to be “Characterization of A π - π Stacking Cocrystal of 4-Nitrophthalonitrile Directed Toward Application in Photocatalysis”.

2) Lines 51-52: “These reactions all suffer from the less variety of acceptor catalysts” – this is not clear. Why is the variety of catalysts important?

Response: The development of novel catalyst manifolds, especially that can participate in as catalytic acceptors in the EDA complex photochemistry, will continue to increase the synthetic potential of these methods. Giving that the reported literature requiring a prefunctionalization step, therefore, designing a general catalytic acceptor achieve *in situ* active, was in great value. The description might be potentially misleading, we have revised it as “The EDA complex photochemistry is still highly desired to develop protocols using more varieties of acceptor catalysts as well as electron donor substrates (beyond arylamines or R-RA) in order to achieve diverse transformations”

3) Lines 58-60: “Firstly, the cyano group of the dicyanobenzene derivatives are known to be altered in excited states, which could change their electronic properties.” – this is not clear. How is the cyano group altered? Why is changing the electronic properties advantageous?

Response: In 2003, Nakata group studied the lowest excited triplet states of 1,2- and 1,4-dicyanobenzenes by low temperature matrix-isolation infrared spectroscopy and DFT calculation (ref. 24). They found that when dicyanobenzene derivatives were excited by irradiation, the corresponding lowest excited triplet, T_1 , would be formed. And the carbon-nitrogen triple bond stretching bands in the S_0 state shift to the low-wavenumber side by 172 and 251 cm^{-1} in the T_1 state for 1,2- and 1,4-dicyanobenzenes, respectively. These findings suggest that the quinoide-type structure similar to that in o- and p-benzoquinones contributes to the T_1 states. It revealed that the character of the triple bond in cyanide changes to the accumulated conjugated system $\text{C}=\text{C}=\text{N}$ under irradiation, at the same time, the unpaired electrons are mainly located on the nitrogen atom and carbon atoms attached cyanide on the benzene ring, these changes would decrease the electron density of the aryl ring, enhancing the electron accepting ability.

To make the description clearer, we have revised it as “Firstly, the cyano group of the dicyanobenzene derivatives are known to be altered to quinoide-type structure in excited states, which could change their electronic properties.” And the corresponding references have been cited properly.

4) Lines 60-61: “In addition, the planar structure is less hindered in the vertical axis, offering great opportunity for the π - π interaction” – this is also not clear. No other acceptors have been discussed in this paragraph, so what is the planar structure less hindered than?

Response: 4-Nitrophthalonitrile has a square-planar structure, which present very small steric hinderance in the vertical axis. To make the description clearer, we have also revised it as “the square-planar structure presents very small steric hinderance in the vertical axis”.

5) Figure 1b: Isolation of a crystalline EDA complex with trimethoxybenzene does not necessarily mean that a similar complex is formed in DMSO. As a result, comparison of the UV/Vis of the solid-state cocrystal with solutions of 4-nitrophthalonitrile and trimethoxybenzene in DMSO is misleading. For accurate comparison, the cocrystal should be dissolved in DMSO (0.01 mM). Similarly, the solid-state EPR does not necessarily translate to solution-phase photochemical reactions.

Response: Thanks to the great concern.

Firstly, the UV-Vis absorption spectra of 4-nitrophthalonitrile, trimethoxybenzene and the cocrystal in DMSO solution (0.01 mM) have been showed below. It clearly showed that once the cocrystal was dissolved in DMSO, the equilibrium disfavored the π - π stacking of 4-nitrophthalonitrile and trimethoxybenzene. However, even in a very diluted solution (0.01 mM), the cocrystal solution

exhibits stronger absorption at the $\lambda = 350 \sim 500$ nm, and the distinct red shift suggests the existence of cocrystal. The UV-Vis absorption spectra of **1a**, **2a**, and **1a/cat1** further confirmed such a conclusion.

On the other hand, following the suggestion, a series of electron paramagnetic resonance (EPR) spectroscopies were examined. Firstly, using the cocrystal of 4-nitrophthalonitrile and 1,3,5-trimethoxybenzene as starting sample, no radical signal was recorded without irradiation. Interestingly, once the sample was irradiated with 390 nm light (ex situ of the magnet), a build-up of a radical signal, assigned as a carbon radical ($g = 2.003$), was observed in the EPR spectrum. When the saturated solution of 4-nitrophthalonitrile and 1,3,5-trimethoxybenzene in DMSO was examined under irradiation, a similar signal could be obtained, however, in a relatively low resolution, probably due to the low concentration of radical species. Delightedly, a much higher resolution signal could be obtained by switching the solvent to DMF. These results indicated that the electron transfer might indeed proceed between 4-nitrophthalonitrile and 1,3,5-trimethoxybenzene to general the carbon radical species under the irradiation of 390 nm light.

6) Table 1: The reaction also gives a significant amount of product in the absence of catalyst (27%).

This result should be discussed, since it means that **cat3-cat5** are not promoting the reaction.

Response: **Cat2-cat5** are indeed not promoting the reaction. For the reason why the product could be formed in the absence of catalyst, we have re-examined UV-Vis spectra of 4-methylanisole (**1a**), benzylidenemalononitrile (**2a**), 4-nitrophthalonitrile (**cat1**), **1a/2a**, and **1a/cat1** with same concentrations of the reaction mixture. The corresponding UV-Vis absorption of **1a/cat1** showcased

the increasing absorption at the $\lambda = 350 \sim 500$ nm, indicating the formed EDA complex possessed the new visible-light absorptions to facilitate the photo-excitation. The solution of **1a** and **cat1** became pale-yellow, further confirming the interaction between each other. Interestingly, the mixture of **1a** and **2a** also presented a relative smaller increasing absorption compared with **1a/cat1**, which could explain why the reaction in the absence of **cat1** could proceed in lower efficiency. These new results have been added in the Figure 3b and discussed in main text. And a comment has been added as “Notably, 27% of the desired product could be formed in the absence of **cat1** (Table S1 in SI), indicating that **cat3-cat5** might be not true catalysts and the reaction of **1a** and **2a** could be driven by photo directly in lower efficiency”.

To be mentioned, the reaction of **1a** with benzylidenemalononitrile **2a** might actually proceed under irradiation of 460 nm light. According to UV-Vis spectra, 4-nitrophthalonitrile (**cat1**) did not absorb 460 nm light, while the EDA complex represent small absorption. After switching the light source to 460 nm LEDs, the C-H alkylation product could be obtained in 20% yield successfully. Without **cat1** under irradiation of 460 nm light, only trace amount of **3aa** could be obtained. These results all supported the EDA complex catalysis of **cat1**.

7) Table 1: The structure of the Eosin Y disodium salt is incorrect. Also, the neutral eosin Y should not be depicted as a spirocycle because this is not the photoactive isomer.

Response: We have revised it as suggestion.

8) Product **3pa** is missing from Table 2.

Response: We have revised it as suggestion.

9) Lines 146-148: “when the substrate bearing both methyl and ethyl groups (**1y**) was explored, the alkylation products could be obtained with a ratio of 1:1.2, indicating that the secondary C(sp³)–H bond functionalization thermodynamic favorable.” – A ratio of 1:1.2 means the reaction is unselective, therefore this discussion of which regioisomer is thermodynamically favored should be removed.

Response: We have removed it as suggestion.

10) Lines 194-195: "In addition, the aerobic oxidation of **3aa** in the presence of methylpiperazine produced the amide **7**, which is known as a M1 antagonist" – a reference is needed for this.

Response: We have revised it as suggestion.

10) I do not understand the UV/Vis data in Figure 3b. The donor is not one of the substrates used in the alkylation chemistry and there are no details about the concentrations or solvent used. Also, the acceptor/donor ratios look wrong. The ratio of acceptor to donor under the optimised reaction conditions is 20:1, so why is a ratio of 1000:1 used for the UV/Vis studies? My conclusion from the UV/vis data is that no significant EDA complex formation occurs and that the acceptor **cat1** is directly excited by 390 nm light.

Response: Thanks for the great concern. However, we cannot fully agree with the conclusion that "no significant EDA complex formation occurs". We have presented our reasons above.

For example, we have re-examined UV-Vis spectra of 4-methylanisole (**1a**), benzylidenemalononitrile (**2a**), 4-nitrophthalonitrile (**cat1**), **1a/2a**, and **1a/cat1** with same concentrations of the reaction mixture. The corresponding UV-Vis absorption of **1a/cat1** showcased the increasing absorption at the $\lambda = 350 \sim 500$ nm, indicating the formed EDA complex possessed the new visible-light absorptions to facilitate the photo-excitation. The solution of **1a** and **cat1** became pale-yellow, further confirming the interaction between each other. These new results have been added in the Figure 3b and discussed in main text.

11) What is the mechanism for the reductive hydrocarboxylation of alkynes? It would be useful to include a proposed mechanism in the SI.

Response: A plausible mechanism of alkynes reductive hydrocarboxylation have been proposed in Figure S19 in SI, which involved a hydrocarboxylation and a hydrogenation.

Supporting information:

12) Check all figure and table numbers in the SI. There are repeat numbers (e.g., 4 x Figure S4) and Table 5 should be re-numbered "Table S3" to match the format of the other tables.

Response: We have revised it as suggestion.

13) Figure S15: It is not clear why the different ratios of **cat1** : HCOOK have been used. It looks like a Job plot is being constructed but no discussion is given.

Response: These experiments were conducted by remaining the same sum amount of **cat1** and HCOOK. And the UV-Vis spectrum of the mixture of HCOOK and 4-nitrophenalonitrile with a ratio of 1:1 shows a strongest red shift, which support the formation of ion pair of HCOOK and 4-nitrophenalonitrile in the reaction mixture. We have added details and discussed the results in the SI.

14) Figure S15: The solvent used for these studies should be stated.

Response: We have revised it as suggestion.

15) FigureS16: The concentrations for these NMR studies should be included. They should ideally match the reaction conditions.

Response: The ¹H NMR of the mixture HCOOK (0.05 M, 0.1 M and 0.2 M) and 4-nitrophenalonitrile (0.1 M) in DMSO was showed in Figure S18, a clear upfield-shift of the ¹H NMR signal was observed with the addition of HCOOK into 4-nitrophenalonitrile, which indicated the interaction of HCOOK and 4-nitrophenalonitrile.

16) Figure S16: Check the ratio of **cat1**: HCOOK in the second spectrum. Based on the trends, the ratio should be reversed.

Response: We have revised it as suggestion.

17) FigureS16: I would not expect such a dramatic change in the chemical shifts for the protons in nitrophthalonitrile upon formation of an EDA complex. The broadening of the signals between 8.4-9 ppm suggests an interaction, however, the well-resolved new signals from 6.4-7.3 ppm could result from a reaction. The dramatic chemical shift change looks like the nitro group could have been converted to an alcohol, giving 3,4-dicyanophenoxide under the basic conditions. Re-isolation of nitrophthalonitrile is required to confirm that no reaction is occurring.

Response: These chemical shifts could be caused by noncovalent interaction instead of nitro group transformation. And it was a great idea to isolate nitrophthalonitrile to confirm such a conclusion. After the ^1H NMR test, 4-nitrophthalonitrile could be indeed isolated in 70% yield by flash chromatography (eluent: petroleum ether/ethyl acetate = 5/1), indicating the 4-nitrophthalonitrile was remained after quenching the EDA complex.

18) P. S73: Product **3pa** does not look clean. There are 6 aliphatic signals when there should only be 4.

Response: **3pa** have been re-purified and updated these data and spectra (see Supporting Information)

19) P. S80-S86: The compound numbers for the structures on these spectra are wrong.

Response: We have revised it as suggestion.

21) P. S81: The structure of the spectrum of **3xa** state 1:1.52 dr but it is not clear from the NMR how this was determined (it looks like a single diastereomer). The dr for this product should also be added to the scheme on P. S20 and to Table 2 in the manuscript.

Response: For the product **3xa**, only one diastereoisomer was obtained, the description has been revised.

22) The dr's are also missing for products **3aaa** and **3aba** in both the SI and manuscript.

Response: only one diastereoisomer of **3aaa** was obtained, and the dr values of **3aba** have been added in both SI and manuscript.

23) P. S104-S107: Check the numbers for the structures on the spectra.

Response: We have checked the numbers of all the compounds and their spectra number.

Re: Reviewer 3

Original comment: In this report Xue et al. present a novel electron acceptor molecule that can participate in photocatalytic processes with suitable electron rich donor substrates. The activation proceeds via the generation of an electron-donor-acceptor (EDA) complex, which can absorb light acting as photocatalyst for various reactions. EDA complexes have been extensively used in photochemistry, but mostly as transient photoactive species generated between reactants or transient intermediates. The use of EDA acceptors as catalysts is somehow limited. In this regard, the use of highly acidic boron species was studied in the past (as correctly mentioned by the authors), and little is known about electron poor aromatic species, with systematic studies performed only by Melchiorre and others (see ref 21) on specific redox active esters. Although presenting some clear scope

limitations (a methoxy group is needed in the acceptor aromatic ring and often highly activated reagents are used, e.g., dicyano Michael acceptors) the novel EDA acceptor photocatalyst presented by the authors presents a clear conceptual interest. In addition, the characterization study, including an EDA crystal structure, is a clear advance in the area.

Therefore, I recommend publication of this report in Nature Communications, pending the following revisions/comments below:

Response: We first thank reviewer 3's for the kind support and all these constructive and insight suggestions in great details.

Original comment: - The narrative of this paper seems "fragmented", presumably due to the choice of presenting first the Giese transformation and mechanism, followed by other transformations which seem "isolated" and poorly described. Perhaps simply rearranging the structure presenting all the transformations first (e.g., the potential of this photocatalyst), followed by a mechanistic study at the end of the paper using the Giese reaction as benchmark process for mechanistic study, would improve the reading.

Response: We have reorganized the manuscript as suggestion, and all the numbers have been revised.

Original comment: - Indoles are usually good in establishing EDA complexes. Would Me-indole work as a substrate for this reaction?

Response: 1,2-Dimethyl-1H-indole and 1,3-dimethyl-1H-indole was employed under standard conditions, unfortunately, failed to afford the desired alkylation product (see Table S4 in SI).

Original comment: - The following sentence is slightly confusing and should be reworded: "The overall quantum yield (Φ) was determined as 0.384 (<1.0), indicating that the radical chain growth pathway was also lowly possibly involved during the transformation." In "indicating that a radical chain propagation is unlikely to be involved in the transformation...."

Response: We have revised it as suggestion.

Original comment: - Table 3: 1d is incorrectly numbered – this should be 1k

Response: We have revised it as suggestion.

Original comment: - Table 3 and line 162: numbering incorrect – electron deficient substrates are 2f-2i but authors write in text 2f-2g.

Response: We have revised it as suggestion.

Original comment: - Line 172-173 and Table 3: the authors should provide an explanation for why 1a is not suitable with substrates 2m and 2n.

Response: The reaction of **1a** with **2j**, **2m** and **2n** afforded lower yields. We assume that the biphenyl, naphthalene and thiophenyl could be competingly interacted with 4-nitrophthalonitrile and prevent the formation of EDA complex of **1a** and 4-nitrophthalonitrile, therefore, the electron-rich substrate **1k** would be more suitable for these transformations. A comment has been added in main text as “The reaction of **1a** with **2j**, **2m** and **2n** afforded lower yields, probably due to the competing interaction of **2j**, **2m** or **2n** with 4-nitrophthalonitrile prevented from the formation of the EDA complex of **1a** and 4-nitrophthalonitrile”.

Recommendations for the supporting information:

Original comment: - The authors claim that the reaction is performed at room temperature, however the picture for the reaction set up shows no fan is used or any cooling mechanism to maintain room temperature while 2 lamps are in operation. The temperature should be measured and presented to facilitate reproducibility.

Response: After measurement by thermometer, the reaction temperature was 29 °C. A comment has been added in SI.

Original comment: - Compounds 3ma, 3pa and 3ua are contaminated with impurities and should be re-purified, and spectra re-recorded in a revised version.

Response: **3ma** and **3pa** have been re-purified and updated these data and spectra (see Supporting Information).

Original comment: - The NMR spectrum of compound 3va is not properly integrated

Response: We have revised it as suggestion.

Original comment: - Finally, ensure the numbering is consistent and correct. Many compounds are numbered incorrectly or have a different number assigned to the NMR spectra.

Response: We have checked the number of all the compounds and their spectra number.

Reviewers' Comments:

Reviewer #1:

Remarks to the Author:

The UV-Vis absorption spectra (Figure 2b) show that the absorption peak is obviously redshifted after the formation of cocrystal, indicating that there is a strong charge transfer interaction between donor and acceptor molecules. Why in the EPR spectrum (Figure 2e), no radical signal was recorded without irradiation.

UV-Vis absorption spectra are used to illustrate the formation of EDA complexes in solution, incomplete spectra is not convincing enough. The formation of EDA complexes in solution should show a more obvious charge transfer absorption peak, rather than a simple redshift on the edge of the absorption peak band.

Reviewer #2:

Remarks to the Author:

The new UV/Vis data in figure 3 provides good support for EDA complex formation. I am satisfied that this new data supports the proposed catalytic EDA mechanism. Therefore, I support publication. However, several of my original points were not fully addressed, so some additional minor revisions should be made first.

1) "To make the description clearer, we have revised it as "Firstly, the cyano group of the dicyanobenzene derivatives are known to be altered to quinoid-type structure in excited states, which could change their electronic properties." And the corresponding references have been cited properly."

This statement is still not clear. The cyano groups cannot have quinoid-type structures, only the dicyanobenzene as a whole can. It is also unnecessary to say that the electronic properties of a molecule are altered in their excited states because this will clearly happen if you change its electronic configuration by exciting it. Most importantly, I do not understand why the electronic structure of the triplet excited state of dicyanobenzene is discussed. This would only be relevant under a photoredox catalysis scenario, whereas the triplet state of dicyanobenzene may not be involved in the proposed EDA complex-mediated reaction. I recommend removing this sentence.

2) "4-Nitrophthalonitrile has a square-planar structure, which present very small steric hinderance in the vertical axis. To make the description clearer, we have also revised it as "the square-planar structure presents very small steric hinderance in the vertical axis".

4-Nitrophthalonitrile does not have a square-planar structure, it has a planar structure as originally stated. Otherwise, the modified sentence is fine.

3) "only one diastereoisomer of 3aaa was obtained, and the dr values of 3aba have been added in both SI and manuscript."

- I think the response should say that the dr for 3aaa has been added and only one diastereomer was observed for 3aba.
- The problem is that four diastereomers should be formed for products 3aaa and 3aba, due to the two stereocenters generated in the reaction and their configuration relative to the distal stereocenters of the sugar and steroid groups. This information needs to be added or at least acknowledged in the SI.
- It is not clear how the dr was determined for 3aaa because the NMR spectrum is not integrated well.

- Also, it is highly unlikely that 3aba could be formed as a single diastereomer when 3sa is formed in 2.2:1 dr. If only a single diastereomer is seen, maybe the other was removed during purification. Crude NMR could be used to confirm this.
- Four stereocenters in the steroid ring are also missing for product 3aba in the SI and substrate 1ab in the manuscript.
- The dr should also be added for 3ya-Et.

Reviewer #3:

Remarks to the Author:

The authors have done a reasonable effort to improve the quality of the manuscript and I believe this manuscript is now ready to be published in Nature Communications.

Re: Reviewer 1

We firstly would like to thank reviewer 1's time and efforts in analyzing our manuscript.

Original comment: The UV-Vis absorption spectra (Figure 2b) show that the absorption peak is obviously redshifted after the formation of cocrystal, indicating that there is a strong charge transfer interaction between donor and acceptor molecules. Why in the EPR spectrum (Figure 2e), no radical signal was recorded without irradiation.

Response: First, UV-Vis spectrophotometry is a laboratory technique used in the measurement of absorbance of light across ultraviolet and visible regions. The absorbance of light causes the transition of molecules from the ground state to the excited state. The redshift of UV-Vis absorption peaks of the co-crystal indeed indicated that the orbitals of donor and acceptor might be degenerated through forming an electron-donor and acceptor (EDA) complex.

Second, the redshift of UV-Vis absorption peaks does not mean that the charge transfer interaction between donor and acceptor would happen spontaneously in ground state. In our case, the charge transfer (electron transfer) did not happen at all in ground state, resulting in recording no radical signal in the dark. Upon irradiation under light, the EDA complex would be excited and an intramolecular electron-transfer (ET) could be proceeded, and a radical intermediate (assigned as a carbon radical ($g = 2.003$)) would be generated and detected by EPR experiment.

Original comment: UV-Vis absorption spectra are used to illustrate the formation of EDA complexes in solution, incomplete spectra is not convincing enough. The formation of EDA complexes in solution should show a more obvious charge transfer absorption peak, rather than a simple redshift on the edge of the absorption peak band.

Response: First, as we mentioned before, a strong and broad absorption peak could be observed in the solid state owing to the highly ordered structure, which could prove

the easy charge transfer between donor and acceptor.

Second, unfortunately, there is only a small redshift observing in solution. This could be explained by the high dispersion of both donor and acceptor in solution, which would result in the much lower efficiency of electron transfer (charge transfer). Although low concentration of donor-acceptor complex formed in solution, such "small" redshift as well as solution color change in the manuscript should be sufficient to prove the formation of EDA complex.

Many other references, such as the relative works reported by Prof. Yao Fu and Paolo Melchiorre (Science 2019, 363, 1429-1434, J. Am. Chem. Soc. 2022, 144, 8914-8919), proved EDA complex while observing only similar "small" redshift of UV-Vis absorption at the $\lambda = 350 \sim 500$ nm in solution.

Re: Reviewer 2

The new UV/Vis data in figure 3 provides good support for EDA complex formation. I am satisfied that this new data supports the proposed catalytic EDA mechanism. Therefore, I support publication. However, several of my original points were not fully addressed, so some additional minor revisions should be made first.

We would like to acknowledge reviewer 2's kind efforts in examining this manuscript and providing all these evaluable and constructive suggestions in great details.

Original comment: 1) "To make the description clearer, we have revised it as "Firstly, the cyano group of the dicyanobenzene derivatives are known to be altered to quinoide-type structure in excited states, which could change their electronic properties." And the corresponding references have been cited properly." This statement is still not clear. The cyano groups cannot have quinoid-type structures, only the dicyanobenzene as a whole can. It is also unnecessary to say that the electronic properties of a molecule are altered in their excited states because this will clearly happen if you change its electronic configuration by exciting it. Most importantly, I do not understand why the electronic structure of the triplet excited state of dicyanobenzene is discussed. This would only be relevant under a photoredox catalysis scenario, whereas the triplet state of dicyanobenzene may not be involved in the proposed EDA complex-mediated reaction. I recommend removing this sentence.

Response: We have removed such a sentence following the suggestion.

Original comment: 2) "4-Nitrophthalonitrile has a square-planar structure, which present very small steric hinderance in the vertical axis. To make the description clearer, we have also revised it as "the square-planar structure presents very small steric hinderance in the vertical axis". 4-Nitrophthalonitrile does not have a square-planar structure, it has a planar structure as originally stated. Otherwise, the modified sentence is fine.

Response: We appreciate this constructive suggestion and have revised it as suggestion.

Original comment: 3) “only one diastereoisomer of **3aaa** was obtained, and the dr values of **3aba** have been added in both SI and manuscript.”

- I think the response should say that the dr for **3aaa** has been added and only one diastereomer was observed for **3aba**. The problem is that four diastereomers should be formed for products **3aaa** and **3aba**, due to the two stereocenters generated in the reaction and their configuration relative to the distal stereocenters of the sugar and steroid groups. This information needs to be added or at least acknowledged in the SI.
- It is not clear how the dr was determined for **3aaa** because the NMR spectrum is not integrated well.
- Also, it is highly unlikely that **3aba** could be formed as a single diastereomer when **3sa** is formed in 2.2:1 dr. If only a single diastereomer is seen, maybe the other was removed during purification. Crude NMR could be used to confirm this.
- Four stereocenters in the steroid ring are also missing for product **3aba** in the SI and substrate **1ab** in the manuscript.

Response: Nice concern.

We have re-examined carefully the ^1H NMR spectrum of compound **3aaa** and found that four diastereomers are indeed formed. Unfortunately, due to the peak overlap and high complexity of the spectra, we cannot determine the accurate ratio of four diastereomers. A footnote e was then added in the manuscript as “Four diastereomers were observed but the ratio was difficult to determine.”

We have also re-conducted and examined the reaction of compound **1ab**, the corresponding ^1H NMR spectrum of the desired product **3aba** showed that four diastereomers were also formed. While peaks of Ar-O-C-H were partially overlapped, based on the integration of the aromatic rings, the ratio of four diastereomers might be 0.10:0.08:1.03:0.90. Notably, we cannot isolate the four pure diastereomers to confirm such a ratio in the reaction, therefore, footnote e as “Four diastereomers were observed but the ratio was difficult to determine” was also added for compound **3aba** in manuscript.

Original comment: The dr should also be added for 3ya-Et.

Response: While compounds **3ya-Me** and **3ya-Et** were not able to isolate using flash chromatography, the ratio was directly determined by ^1H NMR. And notably, only one diastereomer of **3ya-Et** was obtained in the ^1H NMR.

^1H NMR of compound **1y**

¹H NMR of compound 3ya-Me and 3ya-Et

Re: Reviewer 3

The authors have done a reasonable effort to improve the quality of the manuscript and I believe this manuscript is now ready to be published in Nature Communications.

We thank reviewer 3 again for the recognition on this manuscript.

Reviewers' Comments:

Reviewer #1:

Remarks to the Author:

The authors have given reasonable answers to the questions raised, and I agree that the article be published in in Nature Communications.

Reviewer #2:

Remarks to the Author:

(1) Nitrophthalonitrile is not square planar, it is only planar. Please remove "square-".

(2) The four stereocenters in the steroid ring are still missing for product 3aba in the SI and substrate 1ab in the manuscript. See the attached PDF for the structure. Also, the name of the natural product is epiandrosterone, not epirosterone.

CAS RN: 481-29-8
Epiandrosterone

The point-to-point responses:

Manuscript ID: NCOMMS-23-37144B

Title: Characterization of A π - π Stacking Cocrystal of 4-Nitrophthalonitrile Directed Toward Application in Photocatalysis

Author(s): Ting Xue, Cheng Ma, Le Liu, Chunhui Xiao, Shao-Fei Ni, and Rong Zeng

Re: Reviewer 1

The authors have given reasonable answers to the questions raised, and I agree that the article be published in in Nature Communications.

Response: We sincerely appreciate reviewer 1's recognition and efforts on analyzing this manuscript.

Re: Reviewer 2

Original comment: Nitrophthalonitrile is not square planar, it is only planar. Please remove "square-".

Response: We thank reviewer 2's suggestion, we have revised "square-planar structure" to "planar structure".

Original comment: The four stereocenters in the steroid ring are still missing for product **3aba** in the SI and substrate **1ab** in the manuscript. See the attached PDF for the structure. Also, the name of the natural product is epiandrosterone, not epirosterone.

Response: the structures of **3aba** and **1ab** in manuscript and SI have been re-drawn, and the name of natural product have been revised to be "epiandrosterone".